# Machine Learning Techniques for Calorimetry

**Polina Simkina**  **on behalf of the CMS Collaboration**

IRFU, CEA, Université Paris-Saclay, 91190 Gif-sur-Yvette, France; polina.simkina@cern.ch

**Abstract:** The Compact Muon Solenoid (CMS) is one of the general purpose detectors at the CERN Large Hadron Collider (LHC), where the products of proton–proton collisions at the center of mass energy up to 13.6 TeV are reconstructed. The electromagnetic calorimeter (ECAL) is one of the crucial components of the CMS since it reconstructs the energies and positions of electrons and photons. Even though several Machine Learning (ML) algorithms have been already used for calorimetry, with the constant advancement of the field, more and more sophisticated techniques have become available, which can be beneficial for object reconstruction with calorimeters. In this paper, we present two novel ML algorithms for object reconstruction with the ECAL that are based on graph neural networks (GNNs). The new approaches show significant improvements compared to the current algorithms used in CMS.

**Keywords:** machine learning; graph neural network; high energy physics; calorimeter reconstruction

## 1. Introduction

The Compact Muon Solenoid (CMS) experiment [1] is a general-purpose detector at the CERN Large Hadron Collider (LHC). The physics scope of the CMS is to probe the standard model of particle physics and search for the physics beyond the standard model with proton–proton collisions at a center of mass energy from 7 TeV (first collisions in 2010) to 13.6 TeV (collisions recorded since July 2022). In order to do so, it has to be able to efficiently reconstruct the particles coming from these collisions.

Along with the traditional algorithms, the Machine Learning (ML) approach is being broadly implemented, both for event reconstruction and data analysis. Algorithms such as boosted decision trees (BDT) and neural networks (NN) have already been successfully widely applied to the data from Run 2 (e.g., [2,3]). However, more sophisticated algorithms are becoming available, which may bring advantages to the reconstruction techniques in particle physics, using more and more low-level information (e.g., [4,5]).

Graph neural network (GNN) [6–8] is currently one of the most promising ML models. Its main distinguishing characteristics are:

1. GNNs can be applied on the data from complex detector geometries.
2. They are easily applied to sparse data with variable input sizes.
3. GNNs can be applied on non-Euclidean data (unlike convolutional neural networks).
4. In GNNs, the information can flow between close-by nodes of the graph.

In this paper, we will describe two models based on GNNs implemented for the reconstruction of electrons and photons in the CMS electromagnetic calorimeter (ECAL), along with the results achieved by these models and their comparison to the previously used algorithms.

## 2. *e/γ* Reconstruction

Photons and electrons play a crucial role in various physics analyses, including, for example, Higgs boson decays. The reconstruction of the energy and the position of these particles is done using mainly the ECAL. It is also necessary for the measurement of jets' momenta and missing transverse momentum.

*2.1. ECAL*

The ECAL is a homogenous calorimeter made of 75,848 lead tungstate (PbWO$_4$) crystals [9]. It is situated between the tracker and the hadronic calorimeter inside the solenoid, delivering a 3.8 T magnetic field, and is divided into two main parts:

- The barrel with crystal size: 2.2 × 2.2 × 23 cm, covering pseudorapidity $|\eta| < 1.479$.
- The endcaps with crystal size: 2.9 × 2.9 × 22 cm, covering pseudorapidity $1.479 < |\eta| < 3.0$.

*2.2. Reconstruction in the ECAL*

An electron or a photon is reconstructed from the electromagnetic shower in the ECAL. A cluster is built by collecting together the energy deposits (called "rechits") left by this shower in the detector. Each cluster represents a single particle or several overlapping particles. However, electrons and photons can interact with the material in front of the ECAL: electrons emit bremsstrahlung photons, and photons convert into electron–positron pairs, resulting in multiple nearby clusters in the ECAL. These clusters have to be combined to reconstruct the energy of the initial particle. The combination of the sub-clusters is called a SuperCluster [10].

Currently, a geometrical approach is used, called the "Mustache" algorithm. The idea is to combine all the clusters that fall into a specified window around the cluster with the highest energy ("seed") into a SuperCluster. This window has a shape resembling a mustache in the ($\eta$, $\phi$) plane. This shape is chosen because the clusters are wider along the transverse $\phi$-axis rather than the longitudinal $\eta$-axis, due to the CMS magnetic field (3.8 T). The size of the Mustache window depends on the $\eta$-position of the seed and the energy of the cluster.

This algorithm is very efficient; however, there are multiple effects that degrade its performance in terms of energy reconstruction:

- Energy lost before reaching the ECAL, and in detector gaps.
- Energy leakage out of the back of the ECAL.
- The use of finite energy thresholds to suppress noise in the detector electronics.
- Energy deposited by the multiple additional interactions, so-called pileup interactions.

Currently, to mitigate the effect of these issues, a multivariate regression technique (Boosted Decision Tree) trained on simulated photons is used to define an energy correction. The inputs to the BDT are $\approx$30 high-level variables that describe the shower.

Both for the SuperClustering and energy regression tasks, we propose new methods based on state-of-the-art ML tools.

## 3. SuperClustering

*3.1. DeepSC Model*

We developed a new model, called DeepSC, for the SuperClustering. The first step of this algorithm is similar to the Mustache: a window (rectangular shape) is opened around the seed. In the second step, the model predicts whether each cluster in this window belongs to the SuperCluster associated with the corresponding seed, instead of simply taking all of the clusters. Apart from the cluster classification, the DeepSC model also predicts energy correction for each identified SuperCluster [11]. This is the first ML method developed for cluster assignment to the SuperCluster in CMS.

The architecture of the new DeepSC model is presented on Figure 1. The main building blocks of the model are the following:

- Dense layers are used to extract the vectors of the latent features.
- Self-Attention Layers [12,13] that help the network to focus on the most important features.
- Graph Convolutional Network/Graph Highway Network (GHN) [14], where the information can be shared and aggregated between the close-by clusters. The two

algorithms are very similar to each other, with GHN being more robust to over-smoothing during the training.

The architecture of the new DeepSC model, based on GNNs and self-attention layers, is presented in Figure 1.

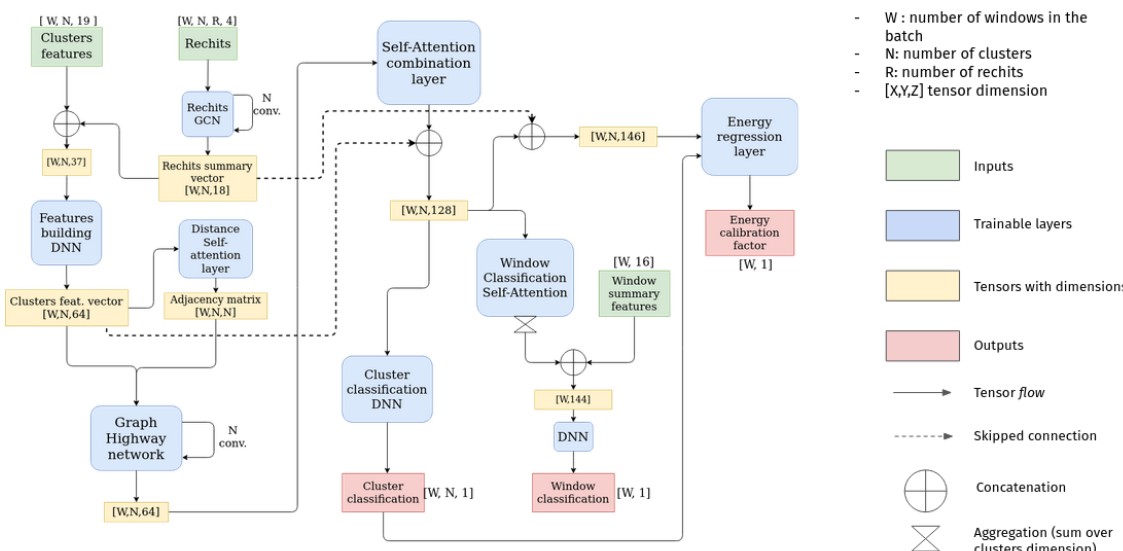

**Figure 1.** DeepSC model architecture. The input to the network consists of selected features and rechits of the clusters that fall into a predefined geometrical window. Using dense layers, the latent features are extracted from the initial input; they are processed and combined together using different types of graph architectures: Graph Convolutional Network (GCN) and Graph Highway Network (GHN). Self-attention layers are used as well, to help the network with focusing on the most important features/inputs. The final outputs are the following: information on whether each of the clusters belongs to the SuperCluster (cluster classification), the type of the particle from which the SuperCluster originated (window classification), and energy correction (energy calibration factor).

In addition to the SuperCluster reconstruction, the model can also predict the flavor of the particle from which the SuperCluster has emerged. The discrimination is done between three flavors: photon, electron, and jet. However, we do not aim at reconstructing the energy of the jets, since it is performed using a standard jet cone algorithm [15]. Therefore, to avoid performance degradation in terms of energy reconstruction for electrons/photons when adding jet discrimination, a ML technique called transfer learning [16] is used. Transfer learning consists of two steps. First, the model is trained only on an electron/photon sample to achieve the optimal performance for the energy reconstruction. Then, the model is re-trained, adding the jet sample, but "freezing" all the parts of the model that are not connected with particle identification. In this way, the reconstruction of the SuperCluster will not be affected by the jet sample.

The DeepSC algorithm is the first attempt to predict the particle flavor, cluster assignment, and energy correction at the same time with ML using raw detector level information.

### 3.2. Dataset Description

A dataset is generated to test the performance of the algorithm. Events are simulated using a full CMS Monte Carlo simulation at 14 TeV, with particles (electrons, photons, and partons) being generated uniformly in pseudorapidity and in a $p_T$ range from 1 to 100 GeV. A pileup scenario with the number of true interactions uniformly distributed in the range of 55 to 75 is used. For the jet sample, every event is required to have at least one photon pair coming from a $\pi_0$.

One entry of the dataset is created in the following way: first, a rectangular window is opened around the seed (an energy threshold of 1 GeV). The size of the window depends

on the position in $\eta$. All the clusters that fall in the specified window around the seed are passed to the network as an input. In more detail, the input for the network contains: cluster information ($E$, $E_T$, $\eta$, $\phi$, z, number of crystals, and information relative to seed: $\Delta\eta$, $\Delta\phi$, $\Delta E$, $\Delta E_T$), list of rechits for each cluster, and summary window features (max, min, mean of $E_T$, $E$, $\Delta\eta$, $\Delta\phi$, $\Delta E$, $\Delta E_T$ of all the clusters in the window).

*3.3. Results*

3.3.1. Energy Resolution

In the case of the DeepSC algorithm, the energy of the initial particle is reconstructed in two steps. First, the energy sum of all the clusters of the network assigned to a SuperCluster is calculated ($E_{Raw}$). Second, the energy correction coefficient is applied to $E_{Raw}$ to achieve a better resolution. In this work, the impact of the first step on energy resolution is presented.

Both Mustache and DeepSC algorithms were applied to the same dataset to compare the performance. Figure 2 shows the resolution of the reconstructed uncorrected SuperCluster energy ($E_{Raw}$) divided by the true energy deposits in ECAL ($E_{Sim}$) versus the transverse energy of the generated particle $E_T^{Gen}$ (left) and the number of simulated pileup (PU) interactions (right). The resolution is computed as being half the difference between the 84% quantile and the 16% quantile (one $\sigma$) of the $E_{Raw}/E_{Sim}$ distribution in each bin. The lower panel shows the ratio of the resolution of the two algorithms: $\sigma_{DeepSC}/\sigma_{Mustache}$. The results are presented for photons; the performance for electrons is similar.

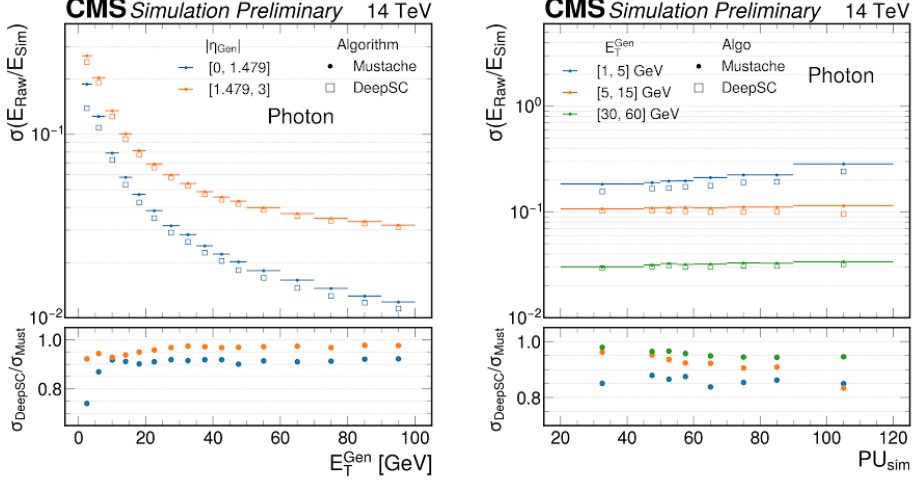

**Figure 2.** Energy resolution for DeepSC and Mustache algorithms. The resolution of the reconstructed uncorrected energy divided by the true energy deposits vs. generated transverse energy of the particle (**left**) and the number of simulated pileup interactions (**right**) is presented. The bottom panels show the ratio of the energy resolutions quantifying the improvement of the DeepSC model over the Mustache algorithm.

The DeepSC algorithm achieves improved performance, especially in the low-$E_T$ and high-pileup regions, where the pileup and the noise significantly degrade the Mustache algorithm resolution. The performance of the DeepSC model in terms of the energy correction results are still under study. In Section 4 of this paper, we discuss another model for energy correction prediction with a similar approach using GNNs on low-level detector information.

3.3.2. Particle Identification

The output of the model for particle identification is the likelihood for the clusters in the window to originate from electron/photon/jet (score). In Figure 3, we show the results obtained for the jet scores in the jet and photon data samples (left), and the electron scores in the photon and electron samples (right).

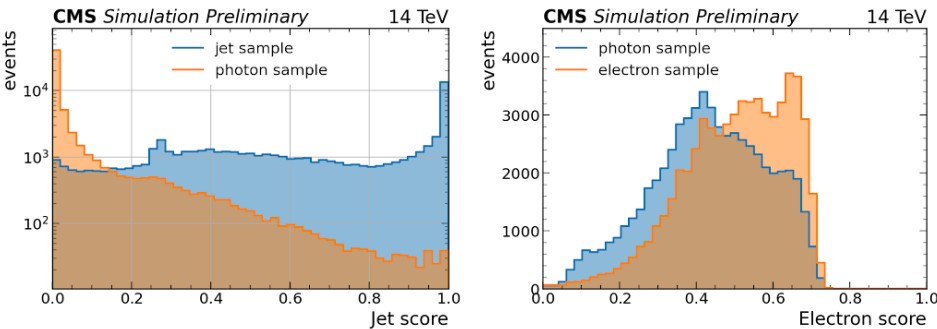

**Figure 3.** Model score distributions for particle identification. The jet score (**left**) represents the likelihood of the SuperCluster to originate from a photon. Clear discrimination between jet and photon samples is visible for this case. The electron score (**right**) represents the likelihood of the SuperCluster to originate from an electron. It demonstrates that despite the absence of the tracker information, some discrimination can be achieved between photon and electron samples.

We can see a clear discrimination between photon and jet samples achieved using the DeepSC model. This information could be used to improve the global CMS event reconstruction (Particle Flow reconstruction [17]), as well as provide extra input information for photon identification algorithms in offline analyses.

The efficient separation between electrons and photons can only be achieved by adding the tracker information to the ECAL. However, it is interesting to see the level of discrimination obtained by the model using only ECAL, and it is worth further investigation.

End-to-end comparison and complementarity with existing algorithms used in CMS are still under study.

## 4. Energy Regression

### 4.1. The Dynamic Reduction Network

For the energy regression task, we also propose to use a neural network. It is the first time raw detector information is used in the ML algorithm for the energy correction in CMS. Generally, neural networks perform best when low-level features are included, and in our case, we use rechits as an input. This will also mitigate the bias coming from human-engineered features, used in the current approach based on a multivariate regression with a BDT. Moreover, the rechits in the calorimeter are quite sparse and vary in number for each particle (from 1 to 100). In this case, it is natural to represent them as points of the graph. Therefore, the new architecture, the dynamic reduction network (DRN) [18] that we have developed for this task, is built on point cloud graph neural network techniques. The input to the model is a point cloud of rechits in the (position, energy) space, and graphs are formed by drawing edges between neighboring hits in a high-dimensional latent space.

The DRN is based on dynamic graph neural networks with the addition of a pooling step analogous to subsampling in CNNs. Our architecture [19] is summarized in Figure 4; the main steps are as follows:

1. The position and energy coordinates of each RecHit are mapped into a high-dimensional latent space using a fully-connected neural network.
2. The message-passing process is performed to aggregate the information between the neighbors and learn the global information.
3. Additional human-engineered features are added to the learned features that were not encoded in the initial hit collection. In particular, two additional features that describe the amount of energy leakage at the back of the ECAL and the energy density from pileup events, are concatenated to the learned features.
4. The resulting set of high-level features is passed through another fully connected neural network to produce the regression output.

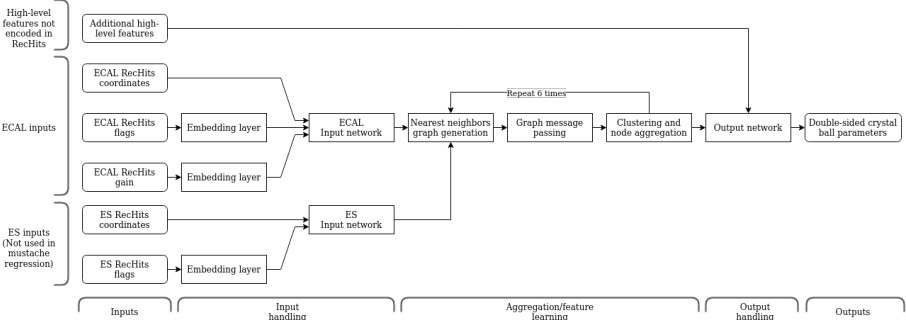

**Figure 4.** Flowchart of the operation of the Dynamic Reduction Network. A point cloud of rechits is mapped into a high-dimensional latent space using a fully-connected neural network, where it is then iteratively transformed and pooled using graph operations. This resulting high-level learned features are then concatenated with extra high-level information not available from the raw collection of rechits, and passed through another fully-connected neural network to obtain the regression output.

### 4.2. Training

The model was trained on realistic detector simulation data, which accurately models particle interactions and detector effects, including pileup. This gives us access to the truth energy values, allowing for supervised training. Our training sample consists of simulated photons with a flat $p_T$ distribution in the range from 25 to 300 GeV fired directly into the detector. Our training data is generated under exactly the same conditions as that used to train the current BDT model.

### 4.3. Results

To compare the performance of the BDT that is currently used in the CMS reconstruction and the DRN model, we applied both of the algorithms to the same photon sample. To obtain the energy resolution, the histograms of $E_{pred}/E_{true}$ were constructed for different transverse momentum ranges $p_T$ and then fitted with the Cruijff function [20] to obtain the key metrics: mean response ($\mu$) and relative resolution ($\sigma/\mu$).

Figure 5 shows the obtained relative resolutions as a function of the particle's transverse momentum $p_T$.

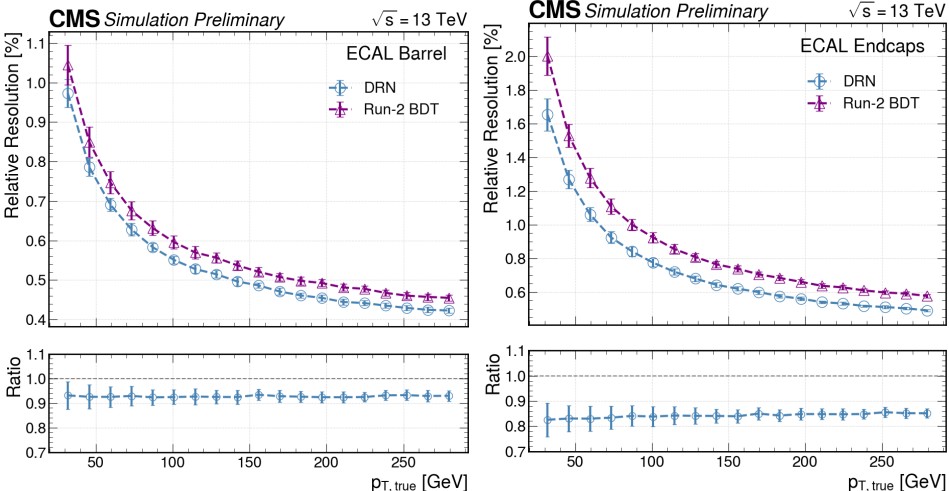

**Figure 5.** Dynamic Reduction Network (DRN) and Boosted Decision Tree performance in the ECAL barrel (**left**) and endcaps (**right**) as a function of generated transverse momentum. The DRN shows an improved resolution by >10%.

The DRN shows an improved resolution by a factor of >10% compared to the BDT for the whole momentum range.

To compare the performance in the actual analysis, the algorithms were also applied on the simulated data for the di-photon invariant mass distributions of H $\rightarrow \gamma\gamma$. The results are shown in Figure 6.

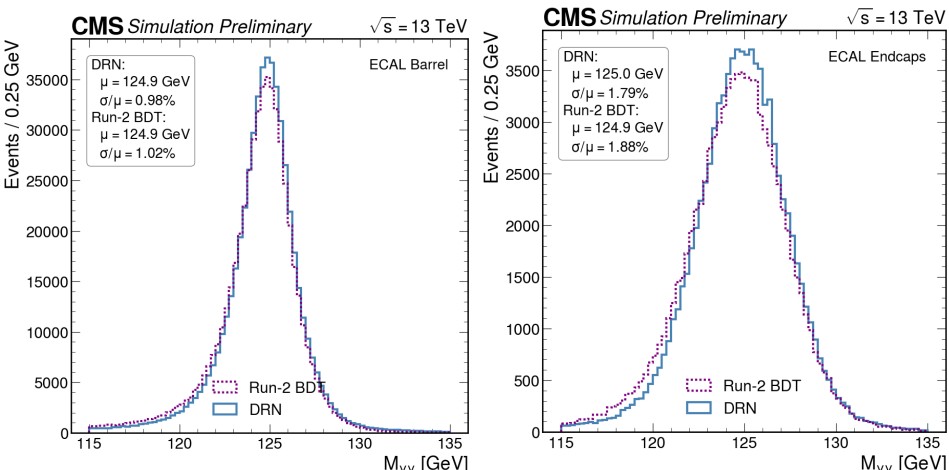

**Figure 6.** Di-photon invariant mass distributions of H $\rightarrow \gamma\gamma$ events for both the Dynamic Reduction Network (DRN) and Boosted Decision Tree architectures in the ECAL barrel (**left**) and endcaps (**right**). The DRN shows an improved resolution by >5% in both detector regions.

In this case, the DRN is able to obtain an improved resolution with respect to the BDT by a factor of >5%, both in the barrel and endcaps of the ECAL.

## 5. Conclusions

In this paper, we presented two novel ML approaches for the reconstruction in calorimetry. Particularly, two different GNN-based architectures were developed for the reconstruction of electromagnetic objects. The DeepSC model can be used for the clustering of energy deposits in the ECAL, as well as to bring extra information on particle identification. The DRN model predicts the energy corrections to be applied to electrons and photons. Both methods show significantly improved performance in energy resolution by about 10 % in comparison to the current reconstruction algorithms used for the ECAL.

Even though the two discussed models are currently developed independently from each other, it is possible to apply them consequently in order to achieve better performance. First, the DeepSC model can be used to retrieve the optimal cluster assignment, and afterwards, the energy correction can be calculated using the DRN. In the future, we plan to combine these two methods for energy reconstruction in the ECAL.

**Funding:** P.S. was supported by the CEA NUMERICS program, which has received funding from the European Union's Horizon 2020 research and innovation program under the Marie Sklodowska-Curie grant agreement No. 800945.

**Data Availability Statement:** Restrictions apply to the availability of these data. Data was obtained from the CMS collaboration and are available with its permission.

**Conflicts of Interest:** The author declares no conflict of interest. The funders had no role in the design of the study; in the collection, analyses, or interpretation of data; in the writing of the manuscript, or in the decision to publish the results.

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
