# Peer review of "Machine Learning Techniques for Calorimetry"

_instruments, doi:10.3390/instruments6040047_

Round 1

Reviewer 1 Report

Thank you for these nice studies on the performance of GNNs for calo reconstruction. The model architectures are interesting and the results appear promising.   In the first work, the model is trained on 3 tasks as I understood it: cluster assignment to supercluster, particle ID, and energy correction estimation. However, the paper only shows results for energy resolution and particle ID. Cluster assignment performance is not shown. Isn't that important? How well does it do?   Though the approaches are interesting, the paper reads like two unrelated works smooshed together and thus lacks some coherence. It would help to have some additional statements comparing them and/or discussing their complementarity.   The model architectures could be better described. Both works use seemingly interesting designs, but most key characteristics (e.g. distance self-attention, graph highway, dynamic reduction) are not discussed at all.   Are these the first ML methods being developed for these tasks? If yes, it should be stated because it gives weight to the work. If not, some mention (and ideally comparison) of previous works is needed.   I am also attaching an annotated pdf of the paper with comments on specific parts.

Author Response

Thank you for your very nice comments! I tried to address all of them the best way I can, here I summarize the changes and answer some of the questions/comments.

In the first work, the model is trained on 3 tasks as I understood it: cluster assignment to supercluster, particle ID, and energy correction estimation. However, the paper only shows results for energy resolution and particle ID. Cluster assignment performance is not shown. Isn't that important? How well does it do?    

  • Indeed there are three tasks that the network is trying to solve. Currently, we are trying to show the performance only of the cluster assignment and the particle ID as the energy correction is still under study. It is true that in the paper it was unclear what is the performance that we are showing for the energy resolution and how it relates to the cluster assignment. I tried to add more details. (L120-124, L135-136)

Though the approaches are interesting, the paper reads like two unrelated works smooshed together and thus lacks some coherence. It would help to have some additional statements comparing them and/or discussing their complementarity. 

  • As mentioned before as the energy correction is not finalized for DeepSC, this stand-alone DRN approach has been developed. On one hand, it gives us confidence that the energy correction prediction is promising, moreover, this standalone approach can be possibly used in Run3 while the DeepSC is being developed. A couple of sentences were added for more relation between the two (L 212-216, L 139-141)

The model architectures could be better described.

  • More details and references were added to the architecture description (L 82-92)

Are these the first ML methods being developed for these tasks?

  • The level of novelty of the algorithms is added. (L80-81, L104-106, L157-158)

Line 18: the wording changed to be more descriptive.

Line 21: references to the analyses, where Machine Learning methods are used, added.

Line 21: wording changed.

Line 24-25: wording changed and more references added.

Line 28: changed to be more precise.

Comment on Line 29: I am not sure I completely understand this comment. While I agree that in the convolutional network the information can flow-by between the pixels of the image, it’s not possible to share the information between two different images (for example). This is what I was trying to point out in this characteristic: the fact that information can be shared between two instances/nodes of the graph (like two different images) using message passing.

Line 69: BDT spelled out.

Comment on Line 85. Concerning the reconstruction of the energy for the jets. The SuperClustering algorithm is really aimed to correct for bremsstrahlung. The energy of the jets is reconstructed with another algorithm (I added the reference for it). Moreover, to reconstruct the energy of the jets, the tracker and the hadronic calorimeter information has to be used as well.

Comment on Line 102. Concerning the clusters falling on the boundaries, they will be treated in the same way as any other cluster in the window, meaning the full cluster will be taken.

Comment on Line 106. Since we are looking at the clusters from bremsstrahlung, the position of the cluster in relation to the seed is quite specific. Even though in principle the network is able to retrieve this information from the given positions of the seed and the cluster, we also add it in the pre-digested form to improve the performance. 

Line 115-116: added the comment that results for the electrons are similar. 

Line 117-118: added comments on the reason why the performance is improved.

Comment on Figure 3. It’s true that the distribution of scores is not the most informative plot in this case. Part of our current study is also to look for better metrics in order to compare the results in the future as well. We have a plot in linear scale but it’s mostly empty. We also have an example of the ROC curve for one energy range, we can add it as an example.

Line 138. Instead of “applying” on the low level features, “including” low level features. 

Line 154-157. Added more explanations on what kind of features are added (extra information).

Comment on consistent notation: it’s true that it’s better to use one notation but we would prefer not to change this for the moment, as all of the plots were already approved by the collaboration in this way.

Line 182. The number for the improvement is cited. 

Reviewer 2 Report

The work presented is very interesting, with good results and discussion. My only general comments is that all figure legends are quite simplistic and could be used to point interesting aspects of the what is being shown. Following are pontual comments to improve the final article: 

Line 24: The listed of distinguishing characteristics of the GNN are quite impressive, but they are not justified. The immediate reference (4) found in the text concludes saying that GNNs   "(...) have not yet been tested in the field with real detector data. (...) It is probably the best practice to start with a simple graph model and architecture then build up on additional complexity geared towards incorporating scientific understanding of the physical process at stake." Please add specific references, or some discussion, to justify the listed claims. 

Figure 1: The legend could be used to point out interesting aspects of the architecture, e.g.: What are the characteristics here that make it more interesting and effective? What is different from non-GNN architectures? Which part of this flow should I be focusing on to understand? 

Line 88: The transfer learning technique is described here in a simplistic way that raises questions about the bias on the final reconstruction. A comment on this, or a reference to some comprehensive study of it, should be added.

Line 114: a simplified version of this description could be part of Figure 2's legend, especially for external use of the figure.

Line 127:  The result on Figure 3 is quite interesting. A quick discussion about how does the final particle id discrimination compares to the other methods, and if they can used simultaneously, would improve the impact of it. 

Author Response

Thank you very much for your comments! I tried to improve the article based on them, most notably, all the figure captions were revised and more references were added to support the claims. 

Line 24: The listed of distinguishing characteristics of the GNN are quite impressive, but they are not justified. The immediate reference (4) found in the text concludes saying that GNNs   "(...) have not yet been tested in the field with real detector data. (...) It is probably the best practice to start with a simple graph model and architecture then build up on additional complexity geared towards incorporating scientific understanding of the physical process at stake." Please add specific references, or some discussion, to justify the listed claims.

  • The references are added to address the specific claims. (Line 24) 

Figure 1: The legend could be used to point out interesting aspects of the architecture, e.g.: What are the characteristics here that make it more interesting and effective? What is different from non-GNN architectures? Which part of this flow should I be focusing on to understand?

  • The legend was largely revised for all the plots. 

Line 88: The transfer learning technique is described here in a simplistic way that raises questions about the bias on the final reconstruction. A comment on this, or a reference to some comprehensive study of it, should be added.

  • I am not quite sure what kind of bias you mean as by construction the energy estimation part is unchanged, thus, there should not be a bias on the energy reconstruction coming from it. However, the reference to transfer learning was missing here (L 98).

Line 127:  The result on Figure 3 is quite interesting. A quick discussion about how does the final particle id discrimination compares to the other methods, and if they can used simultaneously, would improve the impact of it.

  • Unfortunately, we don’t yet have results for the comparison with other algorithms, this is what we are working on at the moment. I added the comment on it (L 153-154)

Round 2

Reviewer 1 Report

I thank the authors for their detailed responses and updates. I find them to be adequate.

Reviewer 2 Report

Thank you for addressing the comments. I have nothing to add.